# Control of single-ligand chemistry on thiolated Au$_{25}$ nanoclusters

Yitao Cao[1], Victor Fung [2], Qiaofeng Yao [1], Tiankai Chen[1], Shuangquan Zang [3], De-en Jiang [2✉] & Jianping Xie [1,4✉]

Diverse methods have been developed to tailor the number of metal atoms in metal nanoclusters, but control of surface ligand number at a given cluster size is rare. Here we demonstrate that reversible addition and elimination of a single surface thiolate ligand (-SR) on gold nanoclusters can be realized, opening the door to precision ligand engineering on atomically precise nanoclusters. We find that oxidative etching of [Au$_{25}$SR$_{18}$]$^-$ nanoclusters adds an excess thiolate ligand and generates a new species, [Au$_{25}$SR$_{19}$]$^0$. The addition reaction can be reversed by CO reduction of [Au$_{25}$SR$_{19}$]$^0$, leading back to [Au$_{25}$SR$_{18}$]$^-$ and eliminating precisely one surface ligand. Intriguingly, we show that the ligand shell of Au$_{25}$ nanoclusters becomes more fragile and rigid after ligand addition. This reversible addition/ elimination reaction of a single surface ligand on gold nanoclusters shows potential to precisely control the number of surface ligands and to explore new ligand space at each nuclearity.

[1] Department of Chemical and Biomolecular Engineering, National University of Singapore, 4 Engineering Drive 4, Singapore 117585, Singapore. [2] Department of Chemistry, University of California, Riverside, CA 92521, USA. [3] Green Catalysis Center, and College of Chemistry, Zhengzhou University, 450001 Zhengzhou, China. [4] Joint School of National University of Singapore and Tianjin University, International Campus of Tianjin University Binhai New City, 350207 Fuzhou, China. ✉email: djiang@ucr.edu; chexiej@nus.edu.sg

Metal nanoclusters have received tremendous research interests in the past two decades due to their atomically-precise structures, intriguing physicochemical properties, and wide potential applications[1–5]. Because of the atomically-precise nature of metal nanoclusters, delicate control of cluster size in synthesis has long been the crucial prerequisite in research[1–6]. Taking thiolate-protected Au nanoclusters as an example, the most traditional and widely-used method is the chemical reduction of Au(I)-thiolate precursors followed by a size-focusing process[7,8]. It has been reported that, in the size-focusing process, oxygen ($O_2$)-generated thiol radicals can induce the selective etching of metastable intermediates by drawing out Au atoms from the metallic core (i.e., top-down process)[9]. As a result, the polydisperse species will transform to thermodynamically stable monodisperse final products.

Thiolate-protected metal nanoclusters are often referred to as $[M_nSR_m]^q$ (SR denotes thiolate ligands), where $n$, $m$, $q$ represent numbers of metal atoms, thiolate ligands and net charge, respectively. The size-focusing will change the $n$ value and result in some thermodynamically stable magic sizes, such as $[Au_{25}SR_{18}]^-$. The high stability of $[Au_{25}SR_{18}]^-$ can be explained by the valence electron counts ($N^* = n - m - q$) of 8, which fulfills the shell closing phenomenon[10]. As a result, $[Au_{25}SR_{18}]^-$ is one of the typical species survived in the size-focusing or ligand-induced etching process[11,12]. Its oxidation will simply change the $q$ value from $-1$ to 0 or $+1$ and introduce slight distortion into its structure, without changing the total molecular formula[13–15].

Tailoring of surface ligands is equally important considering the total synthesis of metal nanoclusters[6]. The surface ligands will greatly influence the cluster size, as well as physicochemical properties, self-assembly behavior, stability and so on[16–27]. Ligand exchange reaction is the main technique used to tailor the surface ligands of thiolate-protected metal nanoclusters[28]. One possible result of a ligand exchange reaction is partial or total substitution of surface ligands without altering the total formula of the parent cluster[29,30]. The other one is the complete change of cluster size, the so-called ligand-exchange-induced size/structure transformation process, which has been applied to discover new cluster species[16,31]. However, all the above methodologies have not addressed an important challenge to precisely control the number of surface ligands ($m$ value) by adding/removing a single ligand at a time for a given number of metal atoms ($n$ value).

To meet the challenge, here we successfully realized the addition and elimination of a surface ligand in thiolate-protected gold nanoclusters based on $[Au_{25}SR_{18}]^-$, the flagship cluster of the $[M_nSR_m]^q$ family. We demonstrated the reversible conversion between water-soluble $[Au_{25}SR_{18}]^-$ and $[Au_{25}SR_{19}]^0$ nanoclusters with identical thiolate ligand through an oxidative etching/reduction cycle (Fig. 1). Detailed evidence from electrospray ionization mass spectrometry (ESI-MS) indicated a continuous oxidation process of the as-prepared $[Au_{25}SR_{18}]^-$ ($N^* = 8$) to $[Au_{25}SR_{18}]^0$ ($N^* = 7$), and then to $[Au_{25}SR_{18}]^+$ ($N^* = 6$). $[Au_{25}SR_{18}]^+$ nanoclusters then reacted with an excess thiolate ligand to form a new isoelectric species, $[Au_{25}SR_{19}]^0$ ($N^* = 6$),

realizing the addition of one thiolate ligand to the original nanocluster. The reverse process can be realized by carbon monoxide (CO)-reduction reaction to eliminate one thiolate ligand in $[Au_{25}SR_{19}]^0$ and regenerate $[Au_{25}SR_{18}]^-$.

## Results

**Synthesis of Au nanoclusters.** $[Au_{25}(MHA)_{18}]^-$ (MHA = 6-mercaptohexanoic acid) was selected as the model cluster in the oxidative etching exploration. The as-prepared sample clearly shows characteristic absorption peaks of $[Au_{25}SR_{18}]^-$ (Fig. 2a, 400, 440, 543, 675, and 800 nm)[32]. Electrospray ionization mass spectrometry (ESI-MS) also indicates the successful synthesis of $[Au_{25}(MHA)_{18}]^-$ species (Fig. 2c). Note that purification was needed before ESI-MS tests, during which the partial oxidation of $[Au_{25}(MHA)_{18}]^-$ already took place. As a result, the species captured in ESI-MS were from a mixture of $[Au_{25}(MHA)_{18}]^-$ and $[Au_{25}(MHA)_{18}]^0$.

**Oxidative etching exploration and product identification.** The as-prepared $[Au_{25}(MHA)_{18}]^-$ nanoclusters were then dialyzed (with a dialysis membrane of molecular weight cut-off of 5000 Da) and the pH was lowered to $9.0 \pm 0.2$ to accelerate the oxidation (see Supplementary Fig. 1). The characteristic absorption peaks of $[Au_{25}MHA_{18}]^-$ disappeared completely within 24 h. Polyacrylamide gel electrophoresis (PAGE) was applied to analyze and purify the final product (Supplementary Fig. 2). The main product was separated for further characterization. UV-Vis absorption spectrum shows significantly blue-shifted absorption band edge compared to $[Au_{25}MHA_{18}]^-$ and characteristic absorption peaks at 440, 545, and 585 nm (Fig. 2b), indicating the formation of a new species. ESI-MS was then applied to determine its molecular formula (Fig. 2d). The isotope pattern only matched well with $[Au_{25}MHA_{19}]^0$, a species with the same nuclearity but different ligand number. Thus, it seems possible to precisely adding one thiolate ligand into $[Au_{25}MHA_{18}]^-$.

**Mechanism of the ligand addition reaction.** To further verify this addition reaction and examine its mechanism, we simplified the whole reaction system by removing the excess ligands, Au(I)-complexes and inorganic ions remained in the solution using PAGE separation. These species may cause the side reactions and the formation of by-products, and a typical dialysis may not be able to completely remove them. The absorption spectrum of the purified $[Au_{25}MHA_{18}]$ species was shown in Fig. 3a.

ESI-MS was then applied to monitor the variation of $[Au_{25}MHA_{18}]$ species after the long-time exposure in air. The overall spectrum indicates the high purity of $[Au_{25}MHA_{18}]$ species (Fig. 3b). However, in a detailed analysis of isotopic patterns (Fig. 3c), we found that the majority of $[Au_{25}MHA_{18}]$ species have been oxidized from $[Au_{25}MHA_{18}]^-$ ($N^* = 8$) to $[Au_{25}MHA_{18}]^0$ ($N^* = 7$). Furthermore, we noticed an excess small peak that can only be assigned to $[Au_{25}MHA_{18}]^+$ ($N^* = 6$), product originated from the further oxidation of $[Au_{25}MHA_{18}]$ species. Actually, it is the first time that the water-soluble $[Au_{25}SR_{18}]$ was found to be oxidized to the valence state of $+1$[13]. As no external oxidant was added to the solution to induce the reaction, the only oxidation resource in this reaction should be $O_2$ in the air. The signals shown in Fig. 3d further support this conjecture. We clearly observed the signals which could be assigned to $[Au_{25}MHA_{18}{}^{\delta+} + O^{\delta-}]$ adducts. This assignment was supported by the fact that these signals become stronger after introducing excess oxidation reagent such as $H_2O_2$ (Supplementary Fig. 3). The $[Au_{25}MHA_{18}{}^{\delta+} + O^{\delta-}]$ adducts should be the intermediate in the reaction between molecular $O_2$ and $[Au_{25}MHA_{18}]$ nanoclusters to generate $[Au_{25}MHA_{18}]^+$.

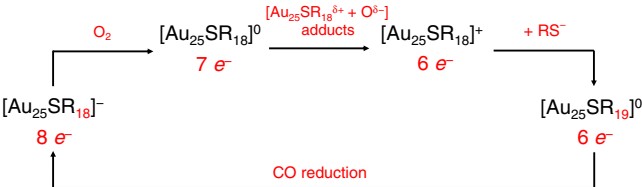

**Fig. 1 Schematic illustration of reaction process.** Schematic illustration of the addition/elimination reaction cycle between $[Au_{25}SR_{18}]^-$ and $[Au_{25}SR_{19}]^0$.

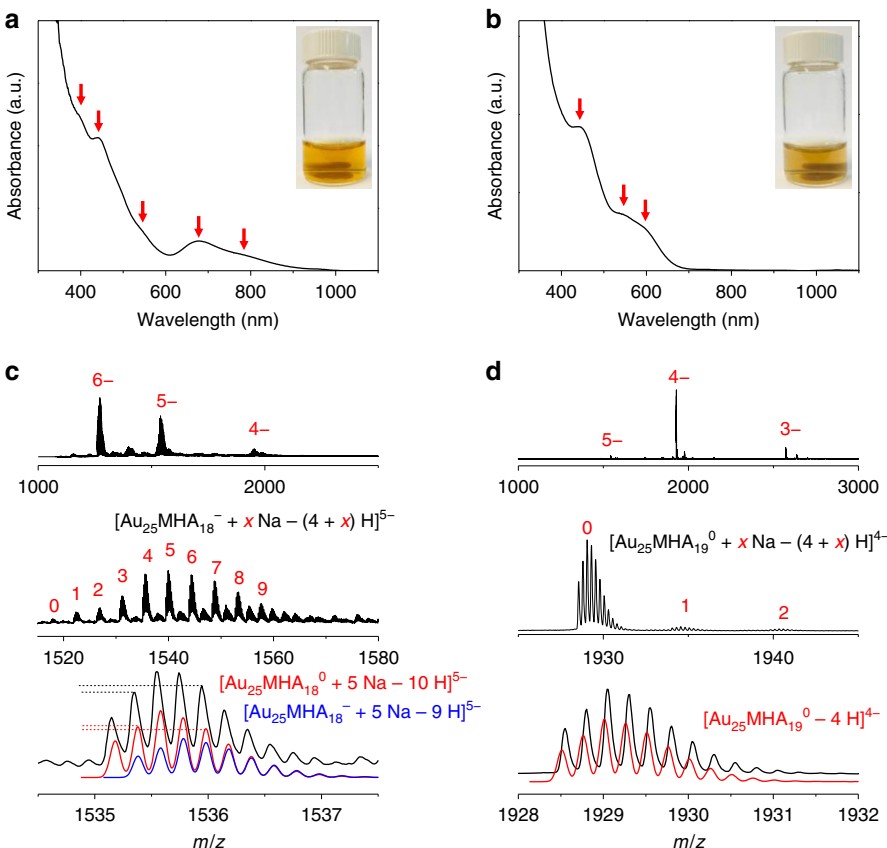

**Fig. 2 Characterization of the Au₂₅ species.** UV-Vis absorption spectra of as-prepared (**a**) $[Au_{25}MHA_{18}]^-$ and (**b**) $[Au_{25}MHA_{19}]^0$. Insets show the digital images of the solution of corresponding species. Red arrows indicate the positions of characteristic absorption peaks. ESI-MS spectra of (**c**) $[Au_{25}MHA_{18}]^-$ and (**d**) $[Au_{25}MHA_{19}]^0$. Both experimental and simulated spectra are shown. Dash lines in **c** are guidelines to judge the relative intensity of corresponding isotopic peaks. The discrepancy between the experimental and simulated result indicate the co-existence of $[Au_{25}MHA_{18}]^-$ and $[Au_{25}MHA_{18}]^0$.

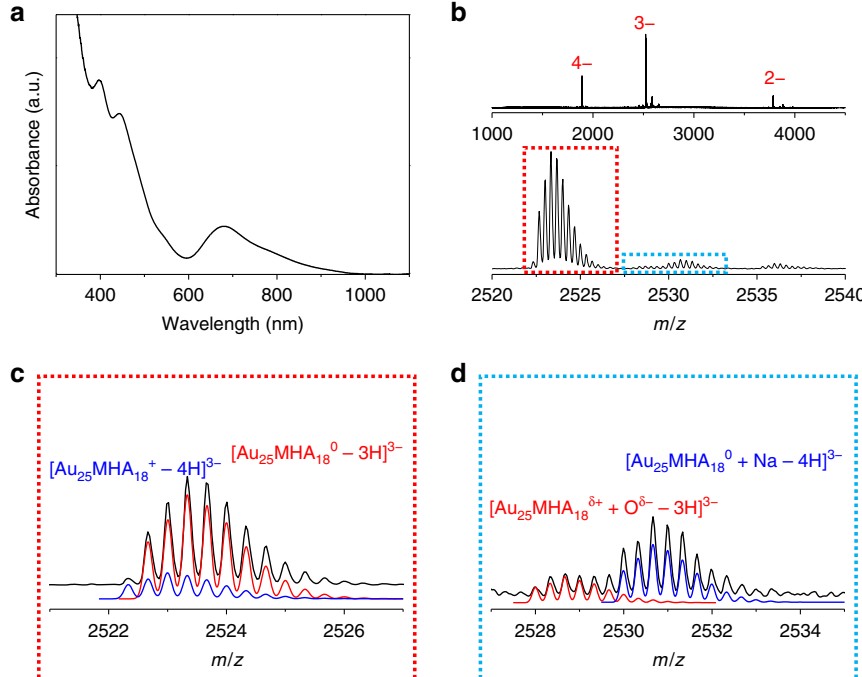

**Fig. 3 Monitoring of reaction intermediates. a** UV-Vis absorption and **b** ESI-MS spectra of $[Au_{25}MHA_{18}]$ species after PAGE purification and long-time exposure in air. Detailed analysis of the enlarged parts in red and blue square is shown in **c** and **d**, respectively. Both experimental and simulated isotopic patterns of labeled species are included.

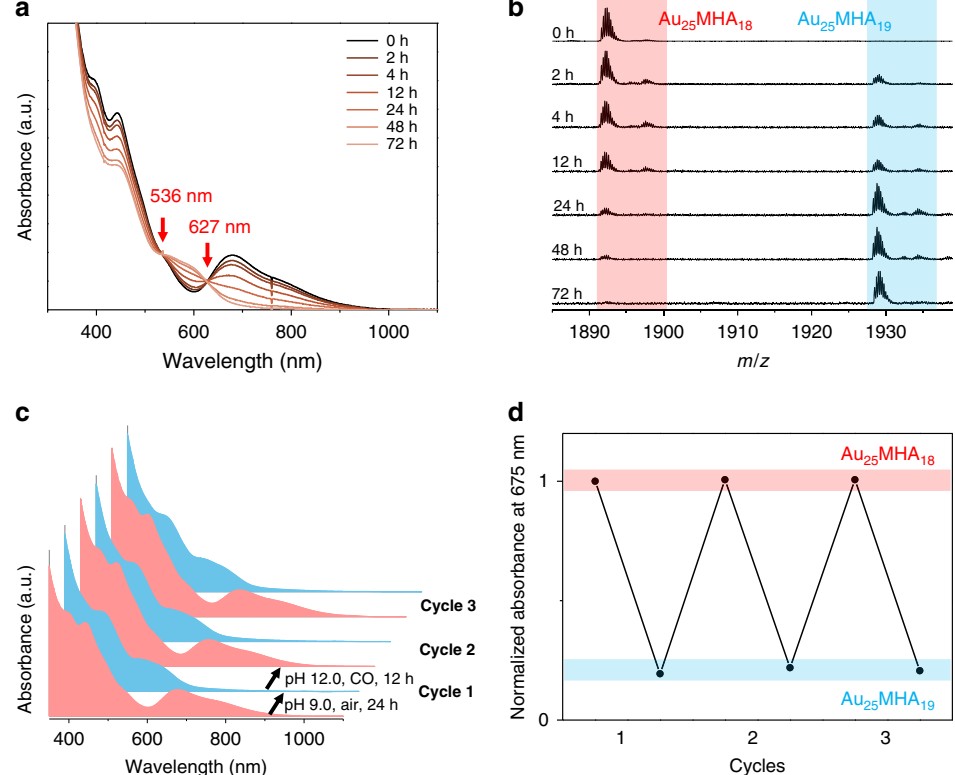

**Fig. 4 Interconversion process. a** Time-course UV-Vis absorption spectra and **b** ESI-MS spectra in the transformation from [$Au_{25}MHA_{18}$] species to [$Au_{25}MHA_{19}$]$^0$ after introducing excess thiol ligands. **c** and **d** show the oxidative etching/reduction cycles between [$Au_{25}MHA_{18}$]$^-$ and [$Au_{25}MHA_{19}$]$^0$ by UV-vis absorption spectra and normalized absorbance at 675 nm in the reversible transformation process, respectively.

Further prolonging the oxidation reaction time will not increase the abundance of [$Au_{25}MHA_{18}$]$^+$ (Supplementary Fig. 4a), indicating an equilibrium between [$Au_{25}MHA_{18}$]$^0$ and [$Au_{25}MHA_{18}$]$^+$ species. However, introducing reducing reagent such as CO and oxidation reagent such as $H_2O_2$ will suppress and increase the signals corresponding to [$Au_{25}MHA_{18}$]$^+$, respectively, further confirming our assignments (Supplementary Fig. 4b and 4c). Since [$Au_{25}MHA_{18}$]$^+$ and final product [$Au_{25}MHA_{19}$]$^0$ have the same $N^*$ value of 6, oxidation is not needed in subsequent transformation process. At this stage, [$Au_{25}MHA_{18}$]$^+$ reacting with one free thiol ligand will drive the isoelectric reaction to the final product [$Au_{25}MHA_{19}$]$^0$.

**Reversible conversion process**. We thus introduced excess thiol ligands into the solution and monitor the transformation process by UV-vis absorption spectroscopy and real-time ESI-MS. We observed two isosbestic points in the UV-vis absorption spectra at 536 and 627 nm (Fig. 4a), which indicates a quasi-one-to-one transformation process (Supplementary Note 1). The ESI-MS spectra (Fig. 4b) also showed a gradual consuming of [$Au_{25}MHA_{18}$] species and generating of [$Au_{25}MHA_{19}$]$^0$ without other observable intermediates. The final product was shown to be pure [$Au_{25}MHA_{19}$]$^0$ nanoclusters (Supplementary Fig. 5). Thus, we have successfully demonstrated that one thiolate ligand can be precisely added to one [$Au_{25}MHA_{18}$]$^-$ to form a [$Au_{25}MHA_{19}$]$^0$.

As many chemical reactions are reversible in nature, we then investigated the possibilities to precisely eliminate one ligand from [$Au_{25}MHA_{19}$]$^0$ to [$Au_{25}MHA_{18}$]$^-$. We were able to realize this transformation by introducing a reducing agent, CO, to [$Au_{25}MHA_{19}$]$^0$ solution. Pure [$Au_{25}MHA_{18}$]$^-$ was formed as indicated by UV-Vis absorption and ESI-MS spectra (Supplementary Fig. 6). Furthermore, the oxidative etching of

[$Au_{25}MHA_{18}$]$^-$ and reduction of [$Au_{25}MHA_{19}$]$^0$ can be reversibly realized (Fig. 4c, d) by simply tailoring the pH of the solution and atmosphere (pH ~9, air for oxidative etching and pH ~12, CO for reduction).

## Discussion

Change of ligand number on the same nuclearity will inevitably influence the structure of ligand shell and thus the properties of whole nanocluster. We try to gain further insights into differences of the ligand shell after addition/elimination of one excess thiolate ligand using tandem mass spectrometry and nuclear magnetic resonance (NMR) analysis. Tandem mass spectrometry has been widely applied in structural analysis of molecular-like clusters and shows sensitivity in analyzing the surface motifs[33,34]. As shown in Fig. 5a, b, the onset energy of fragmentation decreased dramatically from 20 eV of [$Au_{25}MHA_{18}$] species (dominant species: [$Au_{25}MHA_{18}$]$^0$) to 5 eV of [$Au_{25}MHA_{19}$]$^0$, indicating a more fragile nature of the surface structure after addition of one excess thiolate ligand. However, these two species showed a similar fragment pattern that the major fragmentation routes are the same by dissociation of one $Au_4SR_4$ unit stepwise in 1st and 2nd generation fragmentation. In addition, a less prominent 1st generation fragment of [$Au_{25}MHA_{19}$]$^0$ by dissociation of one $Au_5SR_5$ unit has been detected, which should be the result of lengthened motif after ligand addition.

[1]H-NMR signals of surface protecting ligands also provide substantial information of ligand shell in metal nanoclusters[35–37]. As shown in Fig. 6, the signals of protons in MHA surface ligands can be well assigned into two sets in [$Au_{25}MHA_{18}$]$^-$ (see Supplementary Fig. 9 for assignments of the two kinds of surface ligands). Note that $NaBH_4$ was introduced to reduce the [$Au_{25}MHA_{18}$] species to −1 state to get a more analyzable spectrum. In the spectrum of [$Au_{25}MHA_{19}$]$^0$, the most obvious

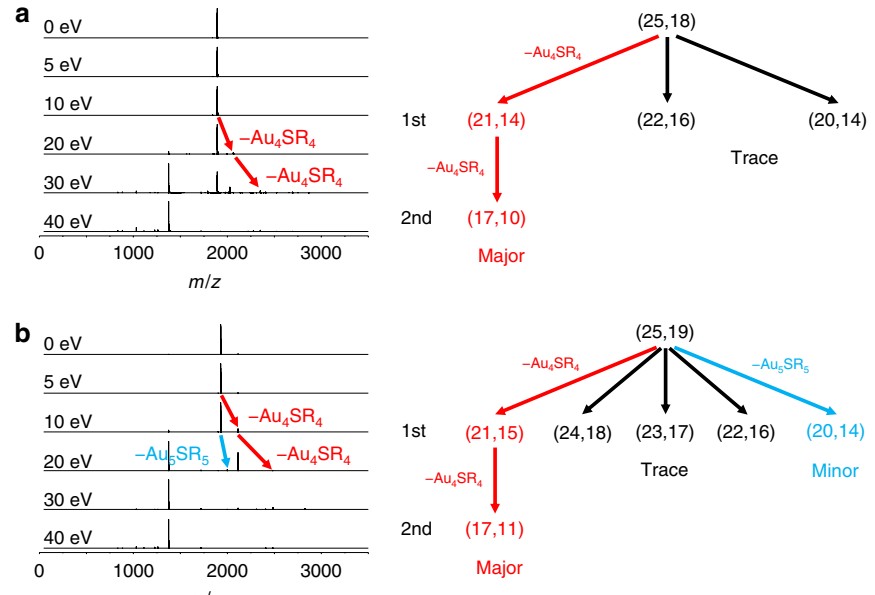

**Fig. 5 Tandem mass testing of the Au$_{25}$ species.** Tandem mass spectra and corresponding fragmentation patterns of (**a**) [Au$_{25}$MHA$_{18}$] and (**b**) [Au$_{25}$MHA$_{19}$]$^0$. ($m$, $n$) denotes species with molecular formula of Au$_m$SR$_n$. Different routes are divided into major, minor, and trace categories based on the abundance in tandem mass spectra. See Supplementary Fig. 7 and 8 for detailed assignments of fragment species.

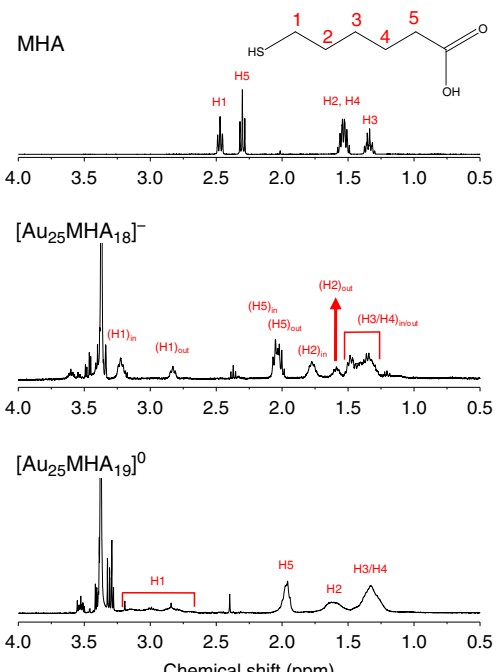

**Fig. 6 $^1$H-NMR testing of the ligand shell.** $^1$H-NMR spectra of MHA ligand, [Au$_{25}$MHA$_{18}$]$^-$, and [Au$_{25}$MHA$_{19}$]$^0$. Sharp peaks around chemical shift of 3.30 and 2.40 ppm are from impurities introduced in PAGE and ultrafiltration process which are hard to be removed and assigned.

differences of the signals were the largely broadened and smoothened peaks. The broadened and featureless signals were the direct result of limited motion of surface ligands[38], indicating a more rigidified ligand shell after addition of one excess thiolate ligand. Note that the peak broadening can also be the result of a distribution of isotropic chemical shift (typically observed in larger metal nanoparticles)[39], a paramagnetic core (observed in metal nanoclusters with odd $N^*$ together with huge down-shift of NMR signals) and electron transfer between different oxidation

states of metal nanoclusters[40], which are not expected to exist in [Au$_{25}$MHA$_{19}$]$^0$.

Summing up the experimental evidence above, we think that the single-ligand chemistry and the interconversion of [Au$_{25}$SR$_{18}$]$^-$ and [Au$_{25}$SR$_{19}$]$^0$ can be described by the following two reactions:

$$[Au_{25}SR_{18}]^- + 1/2\ O_2 + HSR \rightarrow [Au_{25}SR_{19}]^0 + OH^- \tag{1}$$

$$[Au_{25}SR_{19}]^0 + CO + 2OH^- \rightarrow [Au_{25}SR_{18}]^- + HCO_3^- + HSR. \tag{2}$$

Energetics for the two reactions can be computed by density functional theory (DFT) and help understand the thermodynamic driving force of the two reactions. To this end, a structural model for [Au$_{25}$SR$_{19}$]$^0$ would be needed. Given the similar patterns in tandem mass spectra of [Au$_{25}$SR$_{18}$] and [Au$_{25}$SR$_{19}$]$^0$, we hypothesize that they have a similar ligand shell composition. Indeed, we found that the structural model with a defective icosahedral core and lengthened Au$_3$SR$_4$ motif is more stable than many other isomers that we tested (Supplementary Fig. 10; see Supplementary information for computational details). We therefore used this model as our putative structure for [Au$_{25}$SR$_{19}$]$^0$ when computing the reaction energetics. (Here we note that a total structure prediction for [Au$_{25}$SR$_{19}$]$^0$ is still computationally demanding and beyond the scope of this work). The computed ΔE at the B3LYP/TZVP level with an implicit solvation model is −0.11 eV for reaction (1) and −4.57 eV for reaction (2), in qualitative agreement with our experimental observation. In other words, when pH is lowered, a single thiol (HSR) ligand can be oxidatively added to [Au$_{25}$SR$_{18}$]$^-$ under O$_2$ with a favorable energy gain, while [Au$_{25}$SR$_{19}$]$^0$ can be reduced back to [Au$_{25}$SR$_{18}$]$^-$ by CO at higher pH at more favorable energetics.

Besides the favorable energetic of the transformation from [Au$_{25}$SR$_{18}$]$^-$ to [Au$_{25}$SR$_{19}$]$^0$ under the oxidative environment in the presence of free thiol ligands, [Au$_{25}$MHA$_{19}$]$^0$ also shows higher resistance to oxidative etching than [Au$_{25}$MHA$_{18}$]$^-$ (Supplementary Fig. 11). The enhanced stability of [Au$_{25}$MHA$_{19}$]$^0$ under the oxidative environment is probably due to the lengthened motifs and rigidified ligand shell, which will provide better

protection to the Au(0) core and prevent the access of oxidative species (e.g., $O_2$ and thiol radicals). Thus, we were able to obtain the pure product of $[Au_{25}MHA_{19}]^0$ by the as-speculated oxidative etching reaction.

In conclusion, we successfully demonstrated a reaction cycle composed of precise addition and elimination of one thiolate surface ligand on gold nanoclusters. The addition of one thiolate ligand into the original $[Au_{25}MHA_{18}]^-$ nanoclusters was realized by an oxidative etching reaction. In this process, $[Au_{25}MHA_{18}]^-$ was first oxidized by $O_2$ stepwise from $-1$ state to 0 and $+1$ state. The $[Au_{25}MHA_{18}]^+$ then reacted with one free thiol ligand to generate $[Au_{25}MHA_{19}]^0$. Tandem mass spectrometry and $^1$H-NMR indicated that addition of one excess ligand resulted in a ligand shell with similar composition, but more fragile and rigid. We further realized the reverse process by CO-reduction, eliminating one thiolate surface ligand and regenerating $[Au_{25}MHA_{18}]^-$ nanoclusters. Computed reaction energetics from DFT confirms the driving force of the ligand chemistry and the interconversion of $[Au_{25}MHA_{18}]^-$ and $[Au_{25}MHA_{19}]^0$. The key point of such manipulation should be using a stoichiometric amount of ligands in the oxidative etching reaction, which creates a mild reaction condition to possibly stabilize the metastable products. This method provides more opportunities in discovery of 'hidden' species at a given nanocluster size (or metal atom number) with precisely controlled surface ligand number and surface rigidity, and generating new physical and chemical properties, such as photoluminescence and catalysis.

## Methods

**Materials**. 6-mercaptohexanoic acid (MHA), were purchased from Sigma-Aldrich. Hydrogen tetrachloroaurate(III) hydrate ($HAuCl_4 \cdot 3H_2O$) was purchased from Alfa Aesar. Sodium hydroxide (NaOH) was purchased from Sigma-Aldrich. Carbon monoxide (CO, 99.9%) was obtained from Singapore Oxygen Air Liquide Pte Ltd (SOXAL). All chemicals were used without further purification. Ultrapure water (18.2 MΩ cm) was used in all the experiments. All glassware was washed with aqua regia before use.

**Synthesis of $[Au_{25}(MHA)_{18}]^-$ nanoclusters**. $[Au_{25}(MHA)_{18}]^-$ nanoclusters were synthesized through a CO-reduction method. In a typical synthesis, 4 mL of aqueous solution of 5 mM MHA and 0.2 mL of aqueous solution of 50 mM $HAuCl_4$ were added into 5.8 mL of ultrapure water. After 5 min, the pH of the solution was brought up to 12.0 by dropwise adding aqueous solution of 1 M NaOH. After 30 min, CO was bubbled into the solution for 2 min. The reaction vessel was then kept airtight and the reaction was allowed to proceed for 24 h under gentle stirring (500 rpm) at room temperature. The raw product was either purified by ultrafiltration (membrane of molecular weight cut-off = 5000 Da) before testing electrospray ionization mass spectrometry (ESI-MS), or purified by dialysis (membrane of molecular weight cut-off = 5000 Da) before further oxidation reaction.

**Native polyacrylamide gel electrophoresis (PAGE) separation of the Au nanoclusters**. In the native PAGE experiments, Bio-Rad Mini-PROTEAN® Tetra Cell or PROTEAN® II xi Cell system was used. Stacking and resolving gels were prepared by 4 and 30 wt% acrylamide monomers (1:19), respectively. The Au nanocluster solutions were first concentrated by ultrafiltration and then mixed with glycerol to prepare the sample solutions (~10 mM based on Au atoms, 50 vol% glycerol). 2 mL of sample solution were loaded into the well. The PAGE was conducted with constant voltage of 180 V for 24 h at 4 °C. The bands were cut, crushed, and incubated in ultrapure water to obtain solutions of pure Au nanoclusters for further use. The cluster concentration was determined by inductively coupled plasma mass spectrometry (ICP-MS) measurements.

**Conversion from purified $[Au_{25}(MHA)_{18}]$ species to $[Au_{25}(MHA)_{19}]^0$**. Purified $[Au_{25}(MHA)_{18}]$ species from PAGE were used. In a typical reaction, 1 equivalent excess MHA (40 μM) was introduced into the solution of $[Au_{25}(MHA)_{18}]$ (~ 1 mM of [Au]). The solution pH was adjusted to 9.0 ± 0.2. Time-course absorption spectra and ESI-MS spectra were then recorded. Note that the reaction was conducted without stirring in time-course monitoring, and will be accelerated if stirring was applied (completed within 24 h).

**Conversion from $[Au_{25}(MHA)_{19}]^0$ to $[Au_{25}(MHA)_{18}]^-$ by CO reduction**. The solution pH was brought up to 12.0 by an aqueous solution of 1 M NaOH. CO was bubbled into the solution for 2 min. The reaction was proceeded for 12 h airtight. The sample needs to be purified by ultrafiltration before ESI-MS test.

**Characterizations**. Solution pH was recorded by Mettler-Toledo FE 20 pH-meter. UV-vis absorption spectra were recorded by Shimadzu UV-1800 spectrometer. ESI-MS spectra were captured by a Bruker microTOF-Q system in negative ion mode. ESI-MS testing parameters: source temperature 120 °C, dry gas flow rate 8 L per min, nebulizer pressure 3 bar, capillary voltage 3.5 kV, and sample injection rate 3 μL per min. In ESI-MS test, the solution was directly injected without further purification unless specified. $^1$H-NMR spectra were recorded on Bruker 400 MHz system using $D_2O$ as solvent. Purified samples were obtained from PAGE separation and further ultrafiltration process.

## Data availability

All relevant data are available from the corresponding authors on request.

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

## Acknowledgements

We acknowledge the financial support from the Ministry of Education, Singapore, Academic Research Grant R-279-000-580-112 and R-279-000-538-114. DFT computation was supported by the U.S. Department of Energy, Office of Science, Office of Basic Energy Sciences, Chemical Sciences, Geosciences, and Biosciences Division.

## Author contributions

J.X. and Y.C. conceived the idea and designed the experiments. J.X. supervised the project. Y.C. carried out the experiments and characterizations. V.F. and D.J. performed the DFT calculations. Y.C., J.X., and D.J. wrote the manuscript. Q.Y., T.C., and S.Z. discussed the results and commented on the manuscript.

## Competing interests

The authors declare no competing interests.
