## [Peer Review File · Nature Communications]

REVIEWER COMMENTS

Reviewer #1 (Remarks to the Author):

The work by Cao et al. have shown the single ligand level control in Au₂₅ nanoclusters. They found that oxidative etching of [Au₂₅SR₁₈]⁻ leads to an addition of a single ligand and formed [Au₂₅SR₁₉]⁰. The newly formed NCs can be back to Au₂₅SR₁₈ by the CO reduction method. The author demonstrated the mechanistic details of this entire process through the electrospray ionization mass spectrometry. But I find this article to be a very specific and focused study and not important to the general audience of Nature Commun.

The work can be submitted in some specific journals.

However, I do recommend taking care of the following points.

1. The author demonstrated the single ligand addition with only a specific ligand of their choice, namely, 6-mercaptohexanoic acid. It looks like the observation is very specific to MHA. The author does not comment or not even shown such a phenomenon with any other ligands?
2. Why the author chose MHA for this purpose? Is there any specific reason to consider it? Or it's an accidental finding and only specifically observed for MHA ligands?
3. Since it is very specific to MHA, the author should mention this to all places in the manuscript that the NC is Au₂₅(MHA)₁₈. I do agree that previously all NCs were written in such general form Au₂₅(SR)₁₈, but now a day's enormous variety of NCs are reported, and such general formula makes it more confusing to readers about the actual ligand used in work. Only if many ligands were used, such general formula could be written.
4. The NCs transformation goes from 8e to 6e, which is basically an unfavorable reaction in terms of electron count (I'm aware that such transformation is also seen in few cases). What is the driving force in this transformation? Can the author elaborate on that part?
5. How long such Au₂₅(MHA)₁₉ are stable in aerobic conditions?
6. The addition of such extra ligand to Au₂₅(MHA)₁₈ NCs must bring chirality in the NCs due to the structures' anisotropic nature. Can the author prove such properties? A CD spectral report should have been presented.
7. The author can mention the possible application of such ligand controlled NCs to make it more sense that the reader can find an excellent motivation to attempt such control.
8. There are numerous small errors: The author can check thoroughly (e.g., Figure 4a, caption, type in 'wavelength')

9. Figure fronts should be a bit bigger in size; they are hard to read.

10. Figure 2d, the mass spec should be placed a little up from the baseline.

Reviewer #2 (Remarks to the Author):

The authors report the reversible conversion between Au₂₅(SR)₁₈⁻ anion and Au₂₅SR₁₉ nanoclusters, in which R stands for mercaptohexanoic acid. It is remarkable that both are sufficiently stable and uniform to allow the rather comprehensive characterizations. The results are solid and would be of great interest to the broad readership of NatComm. I am in support of publishing with the following suggestions.

Broad impact or relevance: comments on whether they envision the one ligand manipulation is unique to the Au₂₅, or would be applicable to other materials system.

Perhaps the most speculated ligand effects on the noble metal clusters properties is the photoluminescence. combined with the two intermediates, is there more quantitative evidence for the charge transfer mechanism on the quantum efficiency?

The proposed reaction mechanism requires further consideration. If OH⁻ is produced in step 1, how would pH affect the reaction? this seems to contradict some results and discussions unless transition state is viewed differently. Clarification is needed. Related, could there be gold oxides at basic condition that exist as intermediate species? IR or XPS analysis might shed light on these. Further related, MS is destructive, so there is no 'real-time' monitoring of a solution reaction process because the signals are from ionized species in gas phase. If weakly bonded oxygen species or other intermediates did exist, they might not be captured by MS at all.

A minor point is that thiol and thiolate appear to be used interchangeably, scheme, conclusion and a few other places.

Overall, this is a very interesting report that has demonstrated results that are very challenging to harvest.

Reviewer #3 (Remarks to the Author):

The authors first report a reversible addition /elimination reaction of a single surface ligand on gold nanoclusters (Au₂₅SR₁₈ and Au₂₅SR₁₉). This reaction is well-supported by mass spectrometry, UV-Vis and the calculation model. The finding is interesting in a field of metal nanoclusters since the surface ligand control is important for the structures and the physico-chemical properties of gold nanoclusters. My specific comments are described below.

1) How is the long-term stability of Au₂₅SR₁₉ in solution? The 6 electrons in Au₂₅SR₁₉ is likely to be unstable compared to the 8 electrons of Au₂₅SR₁₈⁻.

- 2) Can you explain the reason for the suitable pH values: pH 9, air for oxidation and pH~12, CO reduction?
- 3) How is the photoluminescence spectrum of Au₂₅SR₁₉? The HOMO-LUMO gap of Au₂₅SR₁₉ differs from the case of Au₂₅SR₁₈. The limited motion of surface ligands in Au₂₅SR₁₉ might increase the photoluminescence intensity (QY).
- 4) Can you explain the Au₅SR₅ fragmentation from Au₂₅SR₁₉ in the tandem mass spectra but not for Au₂₅SR₁₈?
- 5) Does the calculation model of Au₂₅SR₁₉ (Fig. S10) reproduce the UV-vis spectrum Fig. 2b)? The calculated geometries of Au₂₅SR₁₉ should be compared with the UV-Vis spectrum of Au₂₅SR₁₉ (Fig. 2b).
- 6) Figure 4a shows a quasi-one-to-one transformation process between Au₂₅SR₁₈ and Au₂₅SR₁₉. According to Figure 1, the intermediate oxidized Au₂₅SR₁₈ should be observed. Why do we observe no intermediate in the UV-Vis spectra ?
- 7) How is the temperature-dependence of the transformation process between Au₂₅SR₁₈ and Au₂₅SR₁₉? Because the etching process generally depends on the reaction temperature.

Replies to reviewers' comments and descriptions of revisions made

Comments by Reviewer #1:

General comments. *The work by Cao et al. have shown the single ligand level control in Au₂₅ nanoclusters. They found that oxidative etching of [Au₂₅SR₁₈]⁻ leads to an addition of a single ligand and formed [Au₂₅SR₁₉]⁰. The newly formed NCs can be back to Au₂₅SR₁₈ by the CO reduction method. The author demonstrated the mechanistic details of this entire process through the electrospray ionization mass spectrometry. But I find this article to be a very specific and focused study and not important to the general audience of Nature Commun. The work can be submitted in some specific journals.*

Reply: We really appreciate the reviewer's efforts in reviewing our manuscript, and providing us constructive comments and suggestions to further improve the quality of our paper. We are sorry that we might not articulate well the significance (to broader readership) and novelty of our work in the previous submission, and we would like to further justify here.

1. Ligand-protected metal nanoclusters have received tremendous research interests due to their atomically-precise structures, intriguing physicochemical properties, and wide potential applications. Because of their atomically precise nature, delicate control of the molecular formulae of final nanocluster products in synthesis has always been the crucial prerequisite in research. Diverse methods have been developed to tailor the number of metal atoms in ligand-protected metal nanoclusters, resulting in a series of cluster sizes. **However, precise control of the number of surface ligands, another important component of ligand-protected metal nanoclusters, at a given cluster size, remains as a grand challenge in the cluster community.** In this paper we report a delicate chemistry to successfully realize the addition and elimination of *a single surface ligand* in thiolate-protected gold nanoclusters (to the best of our knowledge, this is the **first report** in realizing single ligand chemistry in metal nanoclusters), opening the door to precision ligand engineering on atomically precise nanoclusters.
2. The findings revealed in our study show the potential to precisely control the number of surface ligands in ligand-protected metal nanoclusters, which will promote the exploration of new ligand space at each nuclearity of metal nanoclusters. The findings will be of interest to heterogeneous readers of *Nature Communications*, including those in the communities of noble metal chemistry, cluster chemistry, nanochemistry, inorganic chemistry, physical chemistry, and materials chemistry. In addition, this study will also stimulate more intensive fundamental studies on the **precision ligand engineering** (at the atomic and molecular level) for functional metal nanomaterials, further paving their way toward potential applications in diverse fields of applications. It is worth mentioning that **surface engineering represents a forefront research topic in functional nanomaterials.**

3. To further justify the novelty and distinguish our work from the previous studies, we would also like to discuss two key advances of the present study compared to the previous publications, which will also be of interest to readers from diverse communities.

(i) Oxidative etching of gold nanoclusters and oxidation of $[\text{Au}_{25}\text{SR}_{18}]^-$ nanoclusters.

Oxidative etching is crucial in the synthesis of ligand-protected metal nanoclusters and nanoparticles. It is well-demonstrated that the oxidative etching of metal nanoclusters would change the cluster sizes (number of metal atoms). Therefore, the oxidative etching will induce a size-focusing process of metal nanoclusters, leading to the formation of stable nanocluster species (so-called magic-sized nanoclusters), such as $[\text{Au}_{25}\text{SR}_{18}]^-$. In particular, $[\text{Au}_{25}\text{SR}_{18}]^-$ is relatively stable in solution, and the oxidation of this species might also change its valence state from -1 to 0 and +1, without altering their molecular formulae. On the other hand, overoxidation of $[\text{Au}_{25}\text{SR}_{18}]^-$ would finally decompose this species. In contrast, **our work identified a new reaction pathway of metal nanoclusters under the oxidative etching environment, which is addition of a thiolate ligand to generate nanocluster species with the same number of metal atoms but more surface ligands.** To the best of our knowledge, this is the first successful attempt in independently tailoring the number of surface ligands in metal nanoclusters; and we have presented a molecular-level understanding on the underlying chemistry of this reaction pathway.

(ii) Gold nanoclusters with the same nuclearity but different molecular formulae.

There are gold nanoclusters with the same nuclearity but different molecular formulae. **However, the findings revealed in our work are distinctly different.** Most of the cases in the previous publications are realized by ligand exchange reactions. These nanocluster species possess identical Au-S framework but different hydrocarbon tails of thiolate ligands, and they are generally the same kind of species. In this study, we have successfully realized the reversible tailoring of surface ligand number in gold nanoclusters with the same nuclearity and identical surface ligand. **Our protocol shows potential and feasibility in the discovery of new nanocluster species at each nuclearity, which will also be of interest to heterogeneous readers.**

From the above justifications, we have strong confidence that the novelty and significance of our study meet the requirement of *Nature Communications*. In addition, the findings revealed in our study as well as the developed synthetic protocol for precise ligand control, will be of interest to heterogeneous readers from communities of noble metal chemistry, cluster chemistry, nanochemistry, inorganic chemistry, physical chemistry, and materials chemistry. We really appreciate the reviewer's constructive comments/suggestions to further sharpening our mind in this interesting research topic. Thank you.

Comment 1-3. The author demonstrated the single ligand addition with only a specific ligand of their choice, namely, 6-mercaptohexanoic acid. It looks like the observation is very specific to MHA. The author does not comment or not even shown such a phenomenon with any other ligands?

Why the author chose MHA for this purpose? Is there any specific reason to consider it? Or it's an accidental finding and only specifically observed for MHA ligands?

Since it is very specific to MHA, the author should mention this to all places in the manuscript that the NC is $Au_{25}(MHA)_{18}$. I do agree that previously all NCs were written in such general form $Au_{25}(SR)_{18}$, but now a day's enormous variety of NCs are reported, and such general formula makes it more confusing to readers about the actual ligand used in work. Only if many ligands were used, such general formula could be written.

Reply: Thank you for this insightful suggestion. Yes, we used MHA as a model ligand in understanding the transformation chemistry between $Au_{25}SR_{18}$ and $[Au_{25}SR_{19}]^0$, due to the following three reasons. First, MHA is a simple hydrophilic ligand and provides good protection for gold nanoclusters in water. Second, the molecular formulae of water-soluble MHA-protected gold nanoclusters can be readily determined by electrospray ionization mass spectrometry (ESI-MS) without tedious pretreatment, which can facilitate our understandings on the underlying chemistry of the single ligand addition/elimination chemistry. Third, water-soluble MHA-protected gold nanoclusters can also be easily isolated by polyacrylamide gel electrophoresis (PAGE), which can further facilitate our study on the transformation chemistry.

In addition, as suggested by the reviewer, we also obtained preliminary data by using another thiol ligand, 3-mercaptopropionic acid (MPA). As shown in Figure R1a, in the etching product of $Au_{25}MPA_{18}$ nanoclusters, we can clearly observe the absorption features of $[Au_{25}MPA_{19}]^0$ (with impurities; still need further optimization of the reaction conditions), which are similar to those of $[Au_{25}MHA_{19}]^0$. In addition, we have captured the signals of $[Au_{25}MPA_{19}]^0$ by ESI-MS (Figure R1b). This preliminary data suggests that the single ligand chemistry developed in our study is also applicable to other thiol ligand systems with further optimization of the reaction conditions.

Figure R1. (a) UV-Vis absorption spectra of the etching product of $Au_{25}MPA_{18}$ and its comparison with $[Au_{25}MHA_{19}]^0$. The arrows indicate the similar absorption features. (b) ESI-MS spectrum of the etching product of $Au_{25}MPA_{18}$. We have captured the signals of $[Au_{25}MPA_{19}]^0$ with perfect fitting to the calculated isotopic patterns.

We agree with the reviewer that using SR in the entire manuscript will be misleading, and we have changed those SR to MHA where necessary in the section Results and Discussion in the revised manuscript. Thank you for this good suggestion.

Revisions:

Section Results and Section Discussion

SR has been changed to MHA in the manuscript, figures, and Supplementary Information.

Comment 4. *The NCs transformation goes from 8e to 6e, which is basically an unfavorable reaction in terms of electron count (I'm aware that such transformation is also seen in few cases). What is the driving force in this transformation? Can the author elaborate on that part?*

Reply: Thank you for this insightful comment. We agree with the reviewer that gold nanoclusters with a valence electron count of 8 have higher stability than those species with 6 valence electrons. However, the stability of thiolate-protected gold nanoclusters is also dependent on the conditions of the reaction solution. In general, there are two different environments in solution: reductive and oxidative; which will provide the driving forces for the reversible transformation of gold nanoclusters. In particular, under the reductive environment (generally used in most synthetic methods to prepare metal nanoclusters), $[\text{Au}_{25}\text{MHA}_{18}]^-$ has higher stability than $[\text{Au}_{25}\text{MHA}_{19}]^0$. $[\text{Au}_{25}\text{MHA}_{19}]^0$ can be converted back to $[\text{Au}_{25}\text{MHA}_{18}]^-$ in the presence of CO (as a reducing agent) at pH 12, as shown in the reverse transformation process (Fig. 4). On the other hand, under the oxidative environment (in the presence of oxygen), even without the addition of excess MHA ligands, $[\text{Au}_{25}\text{MHA}_{18}]^-$ can be slowly oxidized into 7e species ($[\text{Au}_{25}\text{MHA}_{18}]^0$) and 6e species ($[\text{Au}_{25}\text{MHA}_{18}]^+$), as confirmed by the ESI-MS analysis in Fig. 3. This data suggests that in the presence of O_2 , the 8e species, $[\text{Au}_{25}\text{MHA}_{18}]^-$, is not the most stable species in the solution. The driving force for the transformation from 8e species to 6e species is the oxidation by O_2 in the solution, and the presence of excess thiol ligands can further drive the forward reaction from the 6e $[\text{Au}_{25}\text{MHA}_{18}]^+$ to 6e $[\text{Au}_{25}\text{MHA}_{19}]^0$.

Accordingly, we have provided detailed discussions on this transformation reaction based on our experimental data and DFT calculations. Please refer to Page 12 and 13 for more details.

It should be noted that $[\text{Au}_{25}\text{MHA}_{19}]^0$ is more resistant to decomposition in the oxidative environment than $\text{Au}_{25}\text{MHA}_{18}$ in solution. This also makes possible to obtain the pure product of $[\text{Au}_{25}\text{MHA}_{19}]^0$ after the oxidative etching reaction. We have included a detailed discussion on this in the revised manuscript. Please also refer to our reply to Comment 5 below, regarding the stability of $[\text{Au}_{25}\text{MHA}_{19}]^0$ in solution (Figure R2).

Revisions:

Page 13

“Besides the favorable energetic of the transformation from $[\text{Au}_{25}\text{MHA}_{18}]^-$ to $[\text{Au}_{25}\text{MHA}_{19}]^0$ under the oxidative environment in the presence of free thiol ligands, $[\text{Au}_{25}\text{MHA}_{19}]^0$ also shows

higher resistance to oxidative etching than $[\text{Au}_{25}\text{MHA}_{18}]^-$ (Supplementary Fig. 11). The enhanced stability of $[\text{Au}_{25}\text{MHA}_{19}]^0$ under the oxidative environment is probably due to the lengthened motifs and rigidified ligand shell, which will provide better protection to the Au(0) core and prevent the access of oxidative species (e.g., O_2 and thiol radicals). Thus, we were able to obtain the pure product of $[\text{Au}_{25}\text{MHA}_{19}]^0$ by the as-specified oxidative etching reaction.”

Comment 5. How long such $\text{Au}_{25}(\text{MHA})_{19}$ are stable in aerobic conditions?

Reply: Thank you for this good suggestion. As shown in Figure R2a, $[\text{Au}_{25}\text{MHA}_{19}]^0$ species were stable for at least 2 days in the presence of excess thiol ligands in air. Under the same condition, $\text{Au}_{25}\text{MHA}_{18}$ nanoclusters were easily oxidized and converted to $[\text{Au}_{25}\text{MHA}_{19}]^0$, as demonstrated in our results in Fig. 4. Without excess thiol ligands, $[\text{Au}_{25}\text{MHA}_{19}]^0$ species were stable for at least 7 days in aerobic conditions without observable changes in its absorption spectrum (Figure R2b). Accordingly, we have included these data in the supporting information, and provided a short discussion on the stability of $[\text{Au}_{25}\text{MHA}_{19}]^0$ in the revised manuscript.

Figure R2. (a) UV-Vis absorption spectra of $[\text{Au}_{25}\text{MHA}_{19}]^0$ solution after 2 days in the presence of excess thiol ligands (mole ratio: ligand/cluster = 1/1) in air. (b) UV-Vis absorption spectra of $[\text{Au}_{25}\text{MHA}_{19}]^0$ solution after 7 days in air.

Revisions:

Supplementary Figure 11

Figure R2 has been included as supplementary Fig. 11.

Comment 6. The addition of such extra ligand to $\text{Au}_{25}(\text{MHA})_{18}$ NCs must bring chirality in the NCs due to the structures' anisotropic nature. Can the author prove such properties? A CD spectral report should have been presented.

Reply: This is a very good suggestion, and it will form an interesting research topic in our future study. However, we should apologize that we couldn't address this issue now, as we don't have access to a CD instrument now, and it is difficult for us to reach out for the CD measurement due to

the current situation of COVID-19 in Singapore. We have tried to discuss this issue in the next paragraph and will collect the CD data once the situation for the COVID-19 becomes clearer. We hope the reviewer can understand our difficult situation and consider this as an out-of-scope topic of the current study.

Yes, chirality is a very interesting property of the thiolate-protected metal nanoclusters, and we agree with the reviewer that one excess thiolate ligand on gold nanocluster surface might induce the structural anisotropy and thus chirality of gold nanoclusters. However, as we used the achiral thiol ligand (MHA), although the product might feature chirality, they may exist as a racemic mixture like many other reported thiolate-protected gold nanoclusters using achiral ligand as the protecting ligand, such as Au₁₀₂SR₄₄ (*Science* **2007**, 318, 430) and Au₃₈SR₂₄ (*J. Am. Chem. Soc.* **2010**, 132, 8281). We should thank the reviewer for this good suggestion, and will conduct this accordingly once we have the access to the CD equipment. This will also be a very interesting research topic for a follow up work.

Comment 7. The author can mention the possible application of such ligand controlled NCs to make it more sense that the reader can find an excellent motivation to attempt such control.

Reply: Thank you for this good suggestion. Indeed, we are interested in exploring the potential applications of our product in the field of catalysis. Accordingly, we have included one sentence on this in the revised manuscript.

Revisions:

Page 14

“This method provides more opportunities in discovery of ‘hidden’ species at a given nanocluster size (or metal atom number) with precisely controlled surface ligand number and surface rigidity, and generating new physical and chemical properties, such as photoluminescence and catalysis.”

Comment 8. There are numerous small errors: The author can check thoroughly (e.g., Figure 4a, caption, type in ‘wavelength’)

Reply: We have checked thoroughly the entire manuscript, and made changes accordingly. Thank you.

Comment 9. Figure fronts should be a bit bigger in size; they are hard to read.

Reply: Done. Thank you.

Comment 10. Figure 2d, the mass spec should be placed a little up from the baseline.

Reply: Done. Thank you.

Comments by Reviewer #2:

General Comments: *The authors report the reversible conversion between $Au_{25}(SR)_{18}^-$ anion and $Au_{25}SR_{19}$ nanoclusters, in which R stands for mercaptohexanoic acid. It is remarkable that both are sufficiently stable and uniform to allow the rather comprehensive characterizations. The results are solid and would be of great interest to the broad readership of Nat Comm. I am in support of publishing with the following suggestions.*

Reply: We are thankful for the reviewer's constructive comments/suggestions, and have tried our best to address all the issues raised by the reviewer.

Comment 1. *Broad impact or relevance: comments on whether they envision the one ligand manipulation is unique to the Au_{25} , or would be applicable to other materials system.*

Reply: Thank you for this insightful suggestion. Yes, the single ligand chemistry might be applicable to other types of metal nanoclusters, although we haven't demonstrated its generalized feature in the present study. In previous studies, in a typical size-focusing process of metal nanoclusters driven by the oxidative etching reaction, an excess amount of thiol ligands was normally introduced, generating a relatively harsh reaction condition for metal nanoclusters. Therefore, the metastable nanocluster species would be completely decomposed, and only magic-sized nanocluster species with good stability will survive as the final product. Therefore, the precise manipulation of surface ligands without changing the nanocluster size is almost impossible in such a harsh condition. However, in our study, we used a much milder reaction condition by using a stoichiometric amount of thiol ligands in the oxidative reaction, which will create a reaction condition to possibly stabilize the metastable products. Accordingly, we have included a short sentence on this in the revised manuscript.

Revisions:

Page 14

"The key point of such manipulation should be using a stoichiometric amount of ligands in the oxidative etching reaction, which creates a mild reaction condition to possibly stabilize the metastable products."

Comment 2. *Perhaps the most speculated ligand effects on the noble metal clusters properties is the photoluminescence. combined with the two intermediates, is there more quantitative evidence for the charge transfer mechanism on the quantum efficiency?*

Reply: Thank you very much for this good suggestion. We have measured the photoluminescence properties of both $[Au_{25}MHA_{18}]^-$ and $[Au_{25}MHA_{19}]^0$, and no obvious emission was observed in the range of 480 to 900 nm (the quantum yields of both nanoclusters are very low; please refer to Figure R3). This is a very interesting research topic, and we will further develop strategies to improve the photoluminescence of $[Au_{25}SR_{19}]^0$ in a follow-up work, possibly via the ligand modification of $[Au_{25}SR_{19}]^0$.

Figure R3. Photoemission spectra of $[\text{Au}_{25}\text{MHA}_{18}]^{-}$ and $[\text{Au}_{25}\text{MHA}_{19}]^0$ in the range of 480–900 nm using an excitation wavelength of 460 nm.

Comment 3. *The proposed reaction mechanism requires further consideration. If OH⁻ is produced in step 1, how would pH affect the reaction? this seems to contradict some results and discussions unless transition state is viewed differently. Clarification is needed. Related, could there be gold oxides at basic condition that exist as intermediate species? IR or XPS analysis might shed light on these. Further related, MS is destructive, so there is no 'real-time' monitoring of a solution reaction process because the signals are from ionized species in gas phase. If weakly bonded oxygen species or other intermediates did exist, they might not be captured by MS at all.*

Reply: Thank you for these insightful comments and suggestions. Regarding the pH of the reaction solution, we have optimized the solution pH for the transformation, as shown in Supplementary Figure 1. For the oxidative etching reaction, we need to use a relatively lower pH (pH 9.0), compared to that used in the reductive environment (the reverse transformation reaction in the presence of CO; pH 12.0). We couldn't further reduce the reaction pH as the $\text{Au}_{25}(\text{MHA})_{18}$ nanoclusters were not stable under neutral or acidic pH. The above pH-dependent studies suggest that the oxidative reaction will be accelerated if we decrease the pH of the reaction solution. Therefore, as OH^{-} is produced in step 1, when pH is lowered, a single thiol (HSR) ligand can be oxidatively added to $[\text{Au}_{25}\text{SR}_{18}]^{-}$ under O_2 with a favorable energy gain. OH^{-} will further react with H^{+} from thiol ligands or carboxylic groups on the surface of $\text{Au}_{25}\text{MHA}_{18}$ nanoclusters, which will further drive the reaction toward the formation of $\text{Au}_{25}\text{MHA}_{19}$. Accordingly, we have elaborated this in Discussion section of the DFT calculation (Page 13, lines 8-10).

Regarding the formation of gold oxides, as there are a large amount of thiol ligands in our solution, and thiol has a stronger interaction with Au(I) (to form Au(I)-thiolate bonds) compared to OH^{-} , gold oxides could not be formed in our system. To further obtain the chemical information of the intermediates, instead of XPS, we used ESI-MS to measure the molecular formulae of the intermediates, and found the coordination of O with gold nanoclusters, as shown in Figure 3d. In addition, the only gold source in our reaction is Au_{25} nanoclusters, and we didn't observe the

decomposition of Au₂₅ nanoclusters to form gold oxides (as indicated in Fig. 4), which also suggests the transformation of Au₂₅MHA₁₈ to Au₂₅MHA₁₉ is a quasi-one-to-one process.

Regarding MS: Yes, we agree with the reviewer that some ionization techniques in mass spectrometry are destructive. That is the reason we chose a relatively softer ionization technique, electrospray ionization mass spectrometry (ESI-MS), and we have demonstrated that ESI-MS can be used to capture intermediates and complexes in the reaction solution. For example, we have captured CO adducts with Au₂₅ nanoclusters (*Nat. Commun.* **2017**, *8*, 927). Other groups also used ESI-MS to detect the O₂-adducts of Au nanoclusters (not O atoms) and the complexation between Ag nanoclusters and fullerene (C₆₀) by van-der-Waals forces and π - π interactions (*J. Phys. Chem. C* **2018**, *122*, 19455; *ACS Nano* **2018**, *12*, 2415). Therefore, we believe that our observation and assignment of the O-adducts of Au₂₅ nanoclusters by ESI-MS is reliable. This data is also supported by Supplementary Fig. 3, where the introduction of H₂O₂ in the reaction solution has intensified the corresponding signals of the O-adducts of Au₂₅ nanoclusters.

Thank you for the insightful comments on the above three issues.

Comment 4. *A minor point is that thiol and thiolate appear to be used interchangeably, scheme, conclusion and a few other places.*

Reply: Thank you for the suggestion, and we are sorry for this confusion. We have modified the terms accordingly in the revised manuscript based on the following definition: thiol for H-SR, and thiolate for -SR (most likely bonded to Au(I)).

Revisions:

Page 4

“excess thiolate ligand”

Page 7

“free thiol ligand”

Page 8

“excess thiol ligands”

Page 8

“one thiolate ligand”

Page 13

“free thiol ligand”

General Comments: *Overall, this is a very interesting report that has demonstrated results that are very challenging to harvest.*

Reply: We really appreciate the reviewer’s positive comments on our paper, especially thankful to those constructive suggestions to further improve the readability and quality of our paper. Thank you very much.

Comments by Reviewer #3:

General Comments: The authors first report a reversible addition /elimination reaction of a single surface ligand on gold nanoclusters ($Au_{25}SR_{18}$ and $Au_{25}SR_{19}$). This reaction is well-supported by mass spectrometry, UV-Vis and the calculation model. The finding is interesting in a field of metal nanoclusters since the surface ligand control is important for the structures and the physico-chemical properties of gold nanoclusters. My specific comments are described below.

Reply: We really appreciate the reviewer's constructive comments and suggestions, which have largely improved the quality of our paper. Thank you.

Comment 1. How is the long-term stability of $Au_{25}SR_{19}$ in solution? The 6 electrons in $Au_{25}SR_{19}$ is likely to be unstable compared to the 8 electrons of $Au_{25}SR_{18}^-$.

Reply: Thank you for this insightful suggestion. We have measured the stability of $[Au_{25}MHA_{19}]^0$ in solution. As shown in Figure R4, $[Au_{25}MHA_{19}]^0$ nanoclusters were stable for at least 7 days in air (it will be longer if we further extend the time-course measurement).

Figure R4. UV-Vis absorption spectrum of $[Au_{25}MHA_{19}]^0$ solution (black) and its spectrum after 7 days incubation in air (red).

We agree with the reviewer that gold nanoclusters with a valence electron count of 8 have higher stability than those species with 6 valence electrons. However, the stability of thiolate-protected gold nanoclusters is also dependent on the conditions of the reaction solution. In general, there are two different environments in solution: reductive and oxidative; which will provide the driving forces for the reversible transformation of gold nanoclusters. In particular, under the reductive environment (generally used in most synthetic methods to prepare metal nanoclusters), $[Au_{25}MHA_{18}]^-$ has higher stability than $[Au_{25}MHA_{19}]^0$, and $[Au_{25}MHA_{19}]^0$ will be converted back to $[Au_{25}MHA_{18}]^-$ in the presence of CO (as a reducing agent) at pH 12, as shown in the reverse transformation process (Fig. 4). On the other hand, under the oxidative environment (in the presence of oxygen), even without the

addition of excess MHA ligands, $[\text{Au}_{25}\text{MHA}_{18}]^-$ can be slowly oxidized into 7e species ($[\text{Au}_{25}\text{MHA}_{18}]^0$) and 6e species ($[\text{Au}_{25}\text{MHA}_{18}]^+$), as confirmed by the ESI-MS analysis in Fig. 3. This data suggests that in the presence of O_2 , the 8e species, $[\text{Au}_{25}\text{MHA}_{18}]^-$, is not the most stable species in the solution. The driving force for the transformation from 8e species to 6e species is the oxidation by O_2 in the solution, and the presence of excess thiol ligands can further drive the forward reaction from the 6e $[\text{Au}_{25}\text{MHA}_{18}]^+$ to 6e $[\text{Au}_{25}\text{MHA}_{19}]^0$.

Accordingly, we have provided detailed discussions on this transformation reaction based on our experimental data and DFT calculations. Please refer to Page 12 and 13 for more details.

It should be noted that $[\text{Au}_{25}\text{MHA}_{19}]^0$ is more resistant to decomposition in the oxidative environment than $\text{Au}_{25}\text{MHA}_{18}$ in solution. This also makes possible to obtain the pure product of $[\text{Au}_{25}\text{MHA}_{19}]^0$ after the oxidative etching reaction. We have included a detailed discussion on this in the revised manuscript.

Revisions:

Supplementary Figure 11

Figure R4 has been included in supplementary Fig. 11.

Page 13

“Besides the favorable energetic of the transformation from $[\text{Au}_{25}\text{SR}_{18}]^-$ to $[\text{Au}_{25}\text{SR}_{19}]^0$ under the oxidative environment in the presence of free thiol ligands, $[\text{Au}_{25}\text{MHA}_{19}]^0$ also shows higher resistance to oxidative etching than $[\text{Au}_{25}\text{MHA}_{18}]^-$ (Supplementary Fig. 11). The enhanced stability of $[\text{Au}_{25}\text{MHA}_{19}]^0$ under the oxidative environment is probably due to the lengthened motifs and rigidified ligand shell, which will provide better protection to the Au(0) core and prevent the access of oxidative species (e.g., O_2 and thiol radicals). Thus, we were able to obtain the pure product of $[\text{Au}_{25}\text{MHA}_{19}]^0$ by the as-speculated oxidative etching reaction.”

Comment 2. Can you explain the reason for the suitable pH values: pH 9, air for oxidation and pH~12, CO reduction?

Reply: Regarding the pH of the reaction solution, we have optimized the solution pH for the transformation, as shown in Supplementary Figure 1. For the oxidative etching reaction, we need to use a relatively lower pH (pH 9.0), compared to that used in the reductive environment (the reverse transformation reaction in the presence of CO; pH 12.0). We couldn't further reduce the reaction pH as the $\text{Au}_{25}(\text{MHA})_{18}$ nanoclusters were not stable under neutral or acidic pH. The above pH-dependent studies suggest that the oxidative reaction will be accelerated if we decrease the pH of the reaction solution. Therefore, as OH^- is produced in step 1 (as shown in Page 12, equation 1), when pH is lowered, a single thiol (HSR) ligand can be oxidatively added to $[\text{Au}_{25}\text{SR}_{18}]^-$ under O_2 with a favorable energy gain. OH^- will further react with H^+ from thiol ligands or carboxylic groups on the surface of $\text{Au}_{25}\text{MHA}_{18}$ nanoclusters, which will further drive the reaction toward the formation of $\text{Au}_{25}\text{MHA}_{19}$. While at a higher solution pH (= 12), $[\text{Au}_{25}\text{MHA}_{19}]^0$ can be reduced back

to $[\text{Au}_{25}\text{MHA}_{18}]^-$ by CO with a more favorable energetics (as shown in Page 12, equation 2). Accordingly, we have elaborated this in Discussion section of the DFT calculation.

Comment 3. How is the photoluminescence spectrum of $\text{Au}_{25}\text{SR}_{19}$? The HOMO-LUMO gap of $\text{Au}_{25}\text{SR}_{19}$ differs from the case of $\text{Au}_{25}\text{SR}_{18}$. The limited motion of surface ligands in $\text{Au}_{25}\text{SR}_{19}$ might increase the photoluminescence intensity (QY).

Reply: Thank you very much for this good suggestion. We have measured the photoluminescence properties of both $[\text{Au}_{25}\text{MHA}_{18}]^-$ and $[\text{Au}_{25}\text{MHA}_{19}]^0$, and no obvious emission was observed in the range of 480 to 900 nm (the quantum yields of both nanoclusters are very low; please refer to Figure R5). This is a very interesting research topic, and we will further develop strategies to improve the photoluminescence of $[\text{Au}_{25}\text{SR}_{19}]^0$ in a follow-up work, possibly via the ligand modification of $[\text{Au}_{25}\text{SR}_{19}]^0$.

Figure R5. Photoemission spectra of $[\text{Au}_{25}\text{MHA}_{18}]^-$ and $[\text{Au}_{25}\text{MHA}_{19}]^0$ in the range of 480–900 nm using an excitation wavelength of 460 nm.

Comment 4. Can you explain the Au_5SR_5 fragmentation from $\text{Au}_{25}\text{SR}_{19}$ in the tandem mass spectra but not for $\text{Au}_{25}\text{SR}_{18}$?

Reply: Thank you for this insightful suggestion. The fragmentation of the gold nanoclusters is highly related to their surface motifs (or their intra-cluster rearrangement during the fragmentation). For $\text{Au}_{25}\text{SR}_{18}$, the most common fragmentation is the dissociation of Au_4SR_4 , as demonstrated in Au_{25} nanoclusters protected by different surface ligands, and in different kinds of mass techniques (*J. Am. Chem. Soc.* **2008**, *130*, 5940). The fragmentation of Au_5SR_5 from $\text{Au}_{25}\text{SR}_{18}$ has never been observed in previous reports. The selective dissociation of Au_4SR_4 from $\text{Au}_{25}\text{SR}_{18}$ is due to the intra-cluster rearrangement/fragmentation of surface motifs driven by the unique stability of Au_4SR_4 units, as the surface motifs of $\text{Au}_{25}\text{SR}_{18}$ do not contain such unit. Compared to $\text{Au}_{25}\text{SR}_{18}$, $\text{Au}_{25}\text{SR}_{19}$ has one more thiolate ligand, which feature lengthened motifs, and that is the reason we observed the fragmentation of Au_5SR_5 in $\text{Au}_{25}\text{SR}_{19}$. In addition, Au_5SR_5 has a stable structure similar as Au_4SR_4

units (*J. Am. Chem. Soc.* **2006**, *128*, 10268), and the fragmentation of Au₅SR₅ would occur from the rearrangement process of lengthened surface motifs of Au₂₅SR₁₉.

Comment 5. Does the calculation model of Au₂₅SR₁₉ (Fig. S10) reproduce the UV-vis spectrum Fig. 2b)? The calculated geometries of Au₂₅SR₁₉ should be compared with the UV-Vis spectrum of Au₂₅SR₁₉ (Fig. 2b).

Reply: Thank you for this insightful suggestion. As a total structure prediction of [Au₂₅SR₁₉]⁰ is still computationally demanding, we used a putative structure for [Au₂₅SR₁₉]⁰ to provide the upper bound of the reaction energy to compute the reaction energetics. This putative structure couldn't reproduce the UV-vis absorption spectrum of [Au₂₅SR₁₉]⁰. We have included a short discussion of our DFT calculation, and have provided a short note of "Here we note that a total structure prediction for [Au₂₅SR₁₉]⁰ is still computationally demanding and beyond the scope of the present work".

Comment 6. Figure 4a shows a quasi-one-to-one transformation process between Au₂₅SR₁₈ and Au₂₅SR₁₉. According to Figure 1, the intermediate oxidized Au₂₅SR₁₈ should be observed. Why do we observe no intermediate in the UV-Vis spectra?

Reply: This is another very insightful comment. To understand this phenomenon, we will consider the charge states and corresponding population of [Au₂₅SR₁₈]^q (q = -1, 0, and +1) species during the reaction. As shown in Fig. 3, after a long-time preserving in air without the addition of excess thiol ligands, most of Au₂₅SR₁₈ species are in 0 state with a small amount of nanoclusters of -1 state (Fig. 3a, a shoulder peak at 800 nm) and +1 state (Fig. 3c). [Au₂₅SR₁₈]⁰ is relatively stable in solution and the abundance of [Au₂₅SR₁₈]⁺ didn't increase after a long-time preserving, indicating an equilibrium between [Au₂₅SR₁₈]⁰ and [Au₂₅SR₁₈]⁺ species (Supplementary Fig. 4a). After the introduction of free thiol ligands, [Au₂₅SR₁₈]⁺ will rapidly transform to [Au₂₅SR₁₉]⁰ and the equilibrium will move toward regeneration of the consumed [Au₂₅SR₁₈]⁺. Thus, [Au₂₅SR₁₈]⁺ can be regarded as a reaction intermediate with low concentration. In addition, as these [Au₂₅SR₁₈]^q (q = -1, 0 and +1) species feature very similar absorption spectra (*J. Am. Chem. Soc.* **2007**, *129*, 11322), the influence of small amount of [Au₂₅SR₁₈]⁻ and [Au₂₅SR₁₈]⁺ to the overall absorption spectra of the intermediates during the transformation is marginal. As a result, we observed the changes in the absorption of the reaction solution in Fig. 4a, which indicates the transformation from [Au₂₅SR₁₈]⁰ to [Au₂₅SR₁₉]⁰. This is also the reason for a quasi-one-to-one transformation process. We have provided Supplementary Note 1 to elaborate this.

Revisions:

Supplementary Information, Page 13, Supplementary Note 1

The reply to this comment has been included as Supplementary Note 1.

Supplementary Information, Supplementary Ref. 1

J. Am. Chem. Soc. **129**, 11322 (2007) has been included as supplementary Ref. 1.

Comment 7. How is the temperature-dependence of the transformation process between $Au_{25}SR_{18}$ and $Au_{25}SR_{19}$? Because the etching process generally depends on the reaction temperature.

Reply: We agree with the reviewer that the temperature will influence the transformation process. We have tested the etching process of $Au_{25}SR_{18}$ at 70 °C, and found that $Au_{25}SR_{18}$ decomposed rapidly within 3 hours even without introducing excess thiol ligands (Figure R6). Thus, we believe an elevated temperature will accelerate the etching process of $Au_{25}SR_{18}$ nanoclusters (and completely decompose the parent nanoclusters), which might not be a suitable condition for the single ligand addition reaction revealed in the present study.

Figure R6. UV-Vis absorption spectra of $Au_{25}MHA_{18}$ solution incubated at 70 °C after 3 hours in air.

REVIEWERS' COMMENTS

Reviewer #1 (Remarks to the Author):

Revised version can be accepted in its present form.

Reviewer #3 (Remarks to the Author):

The author response the review's comments and the manuscript was improved for the publication of this journal. I recommend this manuscript for the publication in the present form.